# Exogenous and Endogenous Dendritic Cell-Derived Exosomes: Lessons Learned for Immunotherapy and Disease Pathogenesis

**DOI:** 10.3390/cells11010115

**Published:** 2021-12-30

**Authors:** Mahmoud Elashiry, Ranya Elsayed, Christopher W. Cutler

**Affiliations:** Department of Periodontics, Augusta University, Augusta, GA 30912, USA; MELASHIRY@augusta.edu (M.E.); RELSAYED@augusta.edu (R.E.)

**Keywords:** exosomes, dendritic cells, periodontitis, SASP, immune senescence, *Porphyromonas gingivalis*

## Abstract

Immune therapeutic exosomes, derived exogenously from dendritic cells (DCs), the ‘directors’ of the immune response, are receiving favorable safety and tolerance profiles in phase I and II clinical trials for a growing number of inflammatory and neoplastic diseases. DC-derived exosomes (EXO), the focus of this review, can be custom tailored with immunoregulatory or immunostimulatory molecules for specific immune cell targeting. Moreover, the relative stability, small size and rapid uptake of EXO by recipient immune cells offer intriguing options for therapeutic purposes. This necessitates an in-depth understanding of mechanisms of EXO biogenesis, uptake and routing by recipient immune cells, as well as their in vivo biodistribution. Against this backdrop is recognition of endogenous exosomes, secreted by all cells, the molecular content of which is reflective of the metabolic state of these cells. In this regard, exosome biogenesis and secretion is regulated by cell stressors of chronic inflammation and tumorigenesis, including dysbiotic microbes, reactive oxygen species and DNA damage. Such cell stressors can promote premature senescence in young cells through the senescence associated secretory phenotype (SASP). Pathological exosomes of the SASP amplify inflammatory signaling in stressed cells in an autocrine fashion or promote inflammatory signaling to normal neighboring cells in paracrine, without the requirement of cell-to-cell contact. In summary, we review relevant lessons learned from the use of exogenous DC exosomes for immune therapy, as well as the pathogenic potential of endogenous DC exosomes.

## 1. Introduction

The overall goal of immune therapy is to restore homeostasis or health. Depending on the threat received, immune therapy can consist of various adjuvants or immune checkpoint inhibitors to stimulate tumor destruction or eliminate intracellular pathogens. It can also consist of immune suppressive factors for autoimmune disorders and unresolved chronic inflammatory conditions. The principal antigen presenting cells in the body that direct these pathways (and that are pliable for immune therapy), are dendritic cells (DCs). DCs patrol the peripheral tissues for danger signals from foreign antigens (Ag) and microbes. Internalization that ensues stimulates DC maturation. This entails upregulation chemokine receptors, accessory molecules and inflammatory cytokines needed to initiate an adaptive immune response in the lymphoid organs [1]. For example, mature DCs under the appropriate cytokine milieu direct Ag-specific cytotoxic T cell response for malignant cell destruction [2]. Mature DCs can also direct a T-helper cell 17 (Th17) response to eradicate invading bacteria, though these same effectors, when poorly regulated, can promote degenerative bone diseases [3,4,5]. Immature DCs (iDCs) in contrast can promote immune tolerance by induction of T cell anergy and regulatory T cell (Treg) responses to terminate inflammation [3,6,7,8,9,10] and attenuate inflammatory bone loss [11,12]. The unique capability of DCs to direct these responses has fueled immense interest in their use in immune therapy, for example, by using genetically modified DCs or DCs armed with immunosuppressive cytokines to suppress inflammatory disease in animals. The limitations of such therapy include the rarity of DCs and Treg in vivo and phenotypic instability of DCs [13]. Targeting proinflammatory cytokines using inhibitors of TNF and IL1 has shown therapeutic efficacy for treating inflammatory diseases. However, effects are short lived due to rapid clearance and proteolytic degradation in challenging inflammatory environments [14]. 

Exosomes (EXO), nano-sized extracellular vesicles (EVs) that originate in the endocytic pathway [15], were first discovered from sheep reticulocytes in 1987 [16]. Since then, the ubiquity of exosomes in all tissues and body fluids, including saliva, have been recognized. Intriguingly, EXO can retain sensitive cargo, including proteins, miRNA, mRNA, DNA and lipids, and transfer that cargo to distant sites in the body, where functions of target cells can be modified. This approach has garnered intense interest in the therapeutic potential of EXO [17,18,19,20,21,22,23]. Other than cargo preservation, EXO have many other features of an ideal delivery vehicle, including long circulation time, phenotypic stability and low levels of clearance and degradation. Moreover, EXO released from cells are endowed with surface adhesion molecules and other proteins that promote intercellular communication and uptake by recipient cells. These features are lacking in synthetic agents such as liposomes and other nanoparticles, which can also have toxicity issues [24,25,26,27,28]. EXO can be isolated from DCs, and their small size (30–150 nm) can optimize the delivery of their cargo to recipient cells. Moreover, DC EXO harbor unique proteins that shield them from attack by the complement system [29] and promote binding to tissue integrins [30]. DC EXO can also be loaded with cytokines or peptides using direct or indirect approaches [31,32,33]. EXO from immunostimulatory cargo-loaded mature DCs have been touted for anti-cancer benefits [34], while tolerogenic DC-derived EXO equipped with immunoregulatory cargo offer promise for treatment of autoimmune and inflammatory diseases [35]. The goal of this review is to summarize lessons learned from the use of exogenous DC EXO as immune therapy for chronic diseases, juxtaposed upon the pathogenic potential of endogenously induced DC EXO in infectious/inflammatory diseases.

## 2. Dendritic Cells (DCs): Directors of the Immune Response

DCs consist of multiple subsets, endowed with distinct pattern recognition receptor repertoires, that play distinct functions in the immune system. The phenotypic traits of these subsets, with commonalities and distinctions between mouse and human DC subsets, have been reviewed elsewhere [36]. This review will focus on conventional myeloid DCs, the most numerous DCs in blood and tissues of mice and humans, and which are most active in capture of bacteria. Commensal bacteria fail to activate DC maturation upon phagocytosis [37,38,39,40,41], resulting in an immature phenotype that induces bone protective T-regulatory cells (Tregs) to resolve inflammation [3,6,7,8,9,10]. However, prolonged exposure to dysbiotic bacteria triggers inflammatory signals that drive phenotypic instability, and loss of immune tolerance. DCs emerge in the periphery with hyperinflammatory or mature phenotype, involved in bone damaging Th-17 effector T cells [1,3,4,5,42,43]. 

Immature DCs internalize and process pathogen-associated antigens, stimulating upregulation of proteins needed for activation of efficient antigen presentation to T cells. These signals include MHCI/II molecules (Signal 1), which present processed antigen to T cell receptors (TCR), accessory molecules (e.g., CD86 and CD80) (Signal 2), which bind with CD28 on T cells to amplify signaling needed for optimum T cell activation. Signal 3 consists of cytokines and other soluble proteins that support T cell expansion. Expression of chemokine marker CCR7 by mature DCs guides them to lymph nodes for engagement with naïve T cells [44,45]. A single DC can contact approximately ~5000 T cells within an hour [46]. Free antigens may also travel to the lymph nodes and be presented to T cells by specialized DCs. DCs have the unique capacity to migrate to regional lymph nodes, unlike tissue resident macrophages and initiate clonal expansion of Ag-specific T cells [37].

Instructive in the design of immune therapeutic strategies is the capability of immature or tolerogenic DCs to obtund activation of TCR and CD28 signaling in T cells, resulting in T cell apoptosis or anergy (unresponsiveness) while promoting Treg differentiation. Of particular note is the minimal expression of signals 2 and 3, and high induction of TGFB1 and IL10 in tolerogenic DCs. TGFB1 promotes Treg differentiation while IL10 maintains Treg expansion and phenotypic stability. These cytokines can also inhibit other T and B effector cells and inhibit resident immune cells. In contrast, mature DCs secrete several proinflammatory mediators to promote immunostimulatory T effector cells including Th1, Th2 and Th17 while inhibiting Tregs [10,41]. TGFB1 and IL10 have autocrine functions on DCs, helping to generate and maintain tolerogenic DCs [41,47]. TGFB1 is essential for in vivo development of epidermal Langerhans DCs in mucosa, reported to maintain immune tolerance and mitigate inflammatory bone loss in inflammatory diseases [38,41]. The remarkable plasticity of DCs and other immune cells in an inflammatory environment has led to the development of strategies to reprogram these cells to resolve inflammation through induction of tolerogenic DCs and Tregs [14].

## 3. Exosomes (EXO): The Key to Sustained and Stable Immune Reprogramming 

The discovery of EXO in the late 1980s led to the initial notion that they were waste products result from cellular damage [16]. EXO are derived from the internal vesicles of multivesicular bodies (MVBs) and are released into the extracellular environment by numerous cell types. EXO carry a variety of proteins [48,49,50,51], mRNAs, and small RNAs [51,52], and lipids [51,53,54,55,56] that can influence the phenotype and functions of adjacent cells. EXO can deliver this cargo to target recipient cells locally or at distant sites through release into the bloodstream. The bioactive cargo of EXO, released by all eukaryotic cells, reflects the function and phenotype of donor cells, with the advantage that direct cell-to-cell contact is not necessary for intercellular communication. This has obvious implications for therapeutics and has boosted EXO research in recent years [17,18,19,20,21,22,23].

## 4. General Characteristics of EXO

### 4.1. Size, Morphology and Physical Features

EXO are cup-shaped in scanning electron microscopy (SEM) images, which is an artifact of the fixation/contrast step that induces shrinking of subcellular structures. EXO prepared by Cryo-EM have a round shape. EXO are 30–150 nm in diameter and thus smaller than plasma membrane vesicles (200–1000 nm) [57]. Nanoparticle tracking analysis is used to confirm size distribution of isolated EVs, based on the Brownian motion of vesicles in suspension. Conventional flow cytometry (FC) cannot distinguish between vesicles that are <300 nm, though immunolabeled beads can be used to bind EV ligands, and label them with fluorochrome conjugated antibody to allow visualization by FC [58]. EXO light scatter is correlated to their size, geometry, and composition. EXO also can float in density gradients and have an equilibrium at density of 1.13–1.19 g/mL in sucrose. A large range of densities could reflect the heterogeneity of vesicles from small and large EVs. Subtypes of small EVs could also display different densities [59].

### 4.2. Composition (Protiens, Lipids and Nuclic Acid)

#### 4.2.1. Proteins

EXO contain a specific subset of cellular proteins, some of which depend on the cell type that secretes them, whereas others are found in most EXO regardless of cell type. The latter include proteins from endosomes, the PM, and the cytosol. Proteins from the nucleus, mitochondria, endoplasmic reticulum, and golgi apparatus are deficient in EXO [51]. Two major domains are recognized:

Extracellular Membrane Bound Domain. This domain contains adhesion molecules tetraspanins, integrins, and MFGE8 (lactahedrin). Tetraspanins are highly expressed on EXO membranes and include CD81, CD82, CD9, and CD63. CD63 is more commonly associated in the smallest vesicles (<50 nm), while CD9 is more ubiquitous than other tetraspanins. They facilitate anchoring of multiple proteins and are important for efficient MHCII surface expression and antigen presentation on DCs. They mediate cell adhesion, motility, activation, and proliferation. Several integrins guide EXO biodistribution/trafficking and function. MFGE8 (lactahedrin) can act as an adhesion molecule that binds to phosphatidylserine (PS) exposed on the surface of apoptotic cells and integrins on phagocytic cells to facilitate phagocytosis and clearance of dead cells [48,49,50,51]. Antigen presentation and costimulatory molecules MHCI, MHCII, and CD86 are also in this domain, and commonly derived from antigen presenting cells [48,49,50,51]. Membrane transport/fusion Annexins, flotillins, RABs and ARFs are found here. Annexins act as scaffolding proteins to anchor other proteins to the cell membrane. They are involved in vesicle transport and sorting of endocytic vesicles to early endosomes and fusion of secretory vesicles to plasma membrane for export (endocytosis/exocytosis). Annexins can suppress phospholipases, inhibit arachidonic acid inflammatory metabolites, promote neutrophil detachment, and apoptosis for phagocytosis by macrophages and reduction in recruitment. Flotillins are components of lipid rafts and have a role in endocytosis, trafficking, and cell adhesion. RABs and ARFs are involved in vesicle formation, vesicle movement along actin, and tubulin networks and membrane fusion [48,49,50,51]. Other transmembrane proteins (e.g., LAMPs, TfR) are also described [51]. 

Intracellular Domain of EXO [48,49,50,51] contains ESCRT components (TSG101) and associated proteins (ALIX). TSG101 and ALIX in EXO play roles in multivesicular body (MVB) biogenesis, involving ubiquitin tagged proteins entering endosomes through the formation of intraluminal vesicles. Heat shock protein 70 and 90 are involved in protein folding, clearance of misfolded proteins and stabilization of proteosome and protect cell from stress, while signal transduction molecules include G proteins, syndican, and syntenin. Other cytosolic proteins have been described, such as histones, ribosomal proteins, and proteasome and cytoskeletal proteins, including actin, cofilin, moesin, tubulin, which are expressed. Enzymes such as elongation factors and glyceraldehyde 3-phosphate dehydrogenase are expressed in EXO.

#### 4.2.2. B-Lipids

EXO are enriched with sphingomyelin, phosphatidylserine (PS), cholesterol, and ceramide. This specific lipid composition (e.g., sphingomyelin and cholesterol) resembles detergent-resistant subdomains of the PM called lipid rafts. This is supported by the presence of lipid raft–associated proteins, GPI-anchored proteins and flotillins and resistance of EXO to detergents. It is suggested that lipid rafts could be endocytosed from the plasma membrane (PM) and segregated into intraluminal vesicles (ILVs) during their formation at MVBs and eventually released in EXO [51,53,54,55,56].

#### 4.2.3. Nucleic Acids

mRNAs, miRNAs, short noncoding RNAs and DNA have been identified in EXO cargo. A specific sequence within miRNA has been found to guide sorting to EXO through binding to sumoylated ribonucleoproteins. The protein, heterogeneous nuclear ribonucleoprotein A2B1 (hnRNPA2B1), was shown to recognize and binds specific sequence motifs in miRNAs to facilitate their loading into exosomes. In addition, hnRNPA2B1 sumoylation regulate loading of EXO miRNAs [60]. Another study showed that the RNA binding protein SYNCRIP (synaptotagmin-binding cytoplasmic RNA-interacting protein) directly binds to specific EXO miRNAs sharing common motif [61]. ESCRT-II is an RNA binding complex and may also function to select RNA for incorporation into EXO [51,52]. EXO composition can be changed by modifications of in vitro culture conditions that mimic different extracellular environments or by altering the physiologic, differentiation or maturation state of the secreting cells. For instance, inflammatory signals (e.g., LPS, TNFα, IFNγ) strongly affect the protein and/or RNA composition of EVs released by dendritic, endothelial, or mesenchymal stem cells [51].

## 5. EXO Biogenesis

MVBs and their ILVs can be formed by both ESCRT-dependent and -independent mechanisms, which can lead to distinct EXO.

### 5.1. ESCRT-Dependent Mechanisms

Formation of MVBs and EXO is driven by the endosomal sorting complex required for transport (ESCRT), which is composed of four protein complexes (ESCRT-0, -I, -II and -III) with associated proteins (VPS4, VTA1, ALIX). The ESCRT-0 complex recognizes and sequesters ubiquitinated proteins in the endosomal membrane. ESCRT-0 consists of HRS (hepatocyte growth factor–regulated tyrosine kinase substrate) that recognizes the monoubiquitinated proteins and associates with STAM (signal transducing adaptor molecule). HRS recruits TSG101 of the ESCRT-I complex. ESCRT-I and -II is responsible for endosomal membrane deformation and inward budding with the sorted cargo into the endosomal lumen. ESCRT-I is involved in the recruitment of ESCRT-III via ESCRT-II or ALIX. ESCRT-III induces vesicle scission. The dissociation and recycling of the ESCRT machinery require interaction with the AAA-ATPase VPS4. ESCRT accessory protein Alix is involved in exosome biogenesis and exosomal sorting of syndecans through an interaction with syntenin, which is dependent on ESCRT-II, ESCRT-III, and VPS4 function. ALIX also can bind to transferrin receptors (TfR) and promote TfR sorting onto ILVs of MVBs. Some EV cargo proteins, which are not ubiquitinated, can be also selected. HSC70 chaperone can bind and allow sorting of soluble cytosolic proteins containing a KFERQ sequence to PS on the MVB outer membrane and formation of ILVs in a TSG101- and VPS4-dependent manner. HSC70 can also allow targeting of TfR to EXO. Ubiquitinated and KFERQ-containing proteins are abundant in EXO [49,51].

### 5.2. ESCRT-Independent Mechanisms

Studies show that some cargo is still sorted in EXO after knocking down ESCRT complex proteins through other mechanisms. Tetraspanins CD63, CD81, and CD9 are highly enriched in MVEs and have recently shown to be instrumental in formation ILVs and EXO independently of ESCRT [62,63,64]. Ubiquitinated MHCII (mutant form lacking ubiquitination site that cannot be recognized by ESCRT) can still be sorted to EXO in ESCRT independent manner. MHC class II molecules in EXO are associated with large protein complexes containing tetraspanins, which support their role in EXO biogenesis [51,65], including lipid metabolism enzymes. Neutral sphingomyelinase (nSMase) induces hydrolysis of sphingomyelin into ceramide in the limiting membrane of MVB and induce inward budding and formation of ILVs in an ESCRT-independent manner. NSMase can be inhibited by GW4869 to decrease EXO biogenesis. These observations are consistent with the presence of high concentrations of ceramide in EXO. Phospholipase D2 is another lipid enzyme that allows hydrolysis of phosphatidylcholine into phosphatidic acid and formation of ILVs [55]. A small integral membrane protein of lysosomes and late endosomes (SIMPLE) is secreted in association with EXO and has been shown to also regulate EXO biogenesis [66].

## 6. Mechanisms of EXO Secretion

### 6.1. Role of Rab GTPases in EXO Secretion

Rabs can be involved in vesicle budding, MVB mobility through interaction with the actin or tubulin cytoskeleton (with the help of motors such as myosins and kinesins), tethering, docking, and fusion to the membrane of an acceptor compartment or plasma membrane [67]. RAB11 mediate exocytosis of recycling endosomes [68]. RAB35 mediate exocytosis of early endosomes [69]. RAB27A/B mediate exocytosis of late endosomes [70]. RAB7 also mediate exocytosis of another population of late endosomes. Recycling of early endosomes fuse with the PM due to RAB11 or RAB35, respectively, to secret EXO mainly containing Wnt-associated, Proteolipid protein PLP, TfR, and flotillin. Late endosomes fuse with plasma membrane via RAB27 to release EXO rich in late endosomal proteins including CD63, ALIX, and TSG101. Late endosomes can also fuse with plasma membrane via RAB7 to release EXO rich in ALIX, syntenin-and syndican. Thus, different subtypes of endosomes can generate different population of EXO [51].

### 6.2. Potential Role for Other Molecules in Exosome Secretion

Activation of purinergic receptors by ATP, increases intracellular calcium and cytoskeleton remodeling, promoting EXO secretion. Neurotransmitters depolarize cells promoting EXO exocytosis. Lipopolysaccharide stimulation of T cells, B cell receptor signaling and DC maturation, as well as invasion of DCs by dysbiotic pathogen *Porphyromonas gingivalis* [71] as discussed below, induces release of EXO. Platelets produce more EXO when thrombin receptors are activated. EXO secretion is increased in peptide-loaded immature dendritic upon interaction with T cells. Diacyl glycerol kinase α (DGKα) in T cells has been shown to inhibit EXO secretion while citron kinase is involved in the exocytosis of late endosomal compartments [51].

### 6.3. Role for SNAREs and Other Components of the Fusion Machinery

SNARE proteins form complexes with SNAPs between two membrane compartments and mediate membrane docking and fusion. Ykt6, R SNARE, V0 subunit of V-ATPase, VAMP7, VAMP8, and SNAP-23 are possible mediators of Ca regulated fusion of MVBs with the PM. Different cell types may have distinct SNARE complexes that mediate the fusion of different subpopulations of MVBs within a single cell type. Thus, the downregulation of one SNARE protein might affect the secretion of only a particular subpopulation of EXO [51,72,73,74].

### 6.4. Distinct Populations of MVBs That Modify EXO Secretion

Different subpopulations of morphologically distinct multivesicular bodies (MVBs) based on the size and appearance of the ILVs are known. MVBs bearing early endosomal markers like RAB4 and RAB5A fuse with the PM for EXO release, more commonly than late endosomes. CD63 cholesterol-positive and CD63 lysobisphosphatidic acid (LBPA) negative MVBs are more committed to fusion with the plasma membrane for EXO release, while CD63 cholesterol-negative and CD63 LBPA- positive MVBs are fated for fusion with lysosomes and degradation. In immature DCs, ubiquitinated MHC class II molecules sorted into MVBs mainly are committed for lysosomal degradation. In the presence of antigen-specific T cells, DCs express MHC class II–CD9 complex containing MVBs that fuse with the PM to release EXO. Thus, some MVBs are destined for the degradation pathway, whereas others are fated for exocytosis [51].

## 7. EXO Fate and Mechanism of Uptake in Recipient Cells

In order to understand how EXO may alter physiological and biological functions in vivo, it is essential to study their biodistribution to target organs and cells, and how EXO interact with recipient cells.

### 7.1. Planktonic EXO

Planktonic or free-floating EXO can bind to extracellular matrix and interact with acceptor cells in situ, or they can redistribute through the lymphatics or blood stream and interact with distant cells, tissues, and organs. Circulation of EXO in body fluids is short lived, with ~30 min half-life in plasma after IV administration in lateral tail vein of mice. EXO are typically recovered in lung, liver, and spleen up to 4 h later. EXO can be sequestered by circulating monocyte/macrophages and DCs in the liver and spleen. Moreover, EXO express a variety of integrin and chemokines that can guide in vivo trafficking and tissue homing [75,76,77].

### 7.2. Exosome Cellular Recognition and Target Cell Specificity

Tetraspanins CD63, CD9, and CD81 are instrumental in EXO adhesion, motility, signal transduction and activation of acceptor cells. Differences in tetraspanin complexes influence target cell selection [78,79,80,81,82], spatial assembly of MHCII for antigen recognition and presentation to T cells and may affect the induced TCR signaling [83]. The interaction of ICAM-1 on mature DC-derived EXO with LFA-1 on T cells is critical for efficient naive T cell activation [84]. In addition, several other exosomal integrins have demonstrated roles in adhesion and EXO trafficking [51,75,85,86]. The level of MHC expression on isolated EXO is reflective of their expression on the parent cell. MHC class I molecules on EXO interact with ILT2, ILT4 inhibitory receptors on CD8 plus T cells and NK cells, which promote the inhibition or activation of these cells. EXO from DCs and B lymphocytes express MHC class II for antigen presentation and activation of CD4 T cells [87,88,89,90]. Moreover, EXO can bind and fuse with the plasma membrane of recipient cells, to which they transfer their cargo proteins and RNAs. The phenomenon is regulated by tetraspanin complexes and integrins [86,91].

### 7.3. Soluble and Juxtacrine Signaling by Exosome Surface Ligands

Soluble signaling involves cleavage of ligands from the EXO surface by proteases in the extracellular matrix while juxtracrine signaling requires the juxtaposition or interaction of exosomal surface bound ligands and receptors on target cells. EXO membrane bound FasL, TRAIL and TNF can be cleaved by metalloproteinases to form soluble cytokines. Soluble FasL and TRAIL, TNF and TGFB1 have a reduced activity when compared to that of the membrane-bound form of these ligands [86,92,93,94].

### 7.4. EXO Internalization

Several modes of EXO internalization have been proposed, depending on EXO size and the acceptor cell type. Large EVs or aggregates of small EVs like EXO mostly induce phagocytosis, whereas individual small EVs or EXO can be internalized by nonphagocytic processes, such as pinocytosis [86].

#### 7.4.1. Phagocytosis

Phagocytosis of EXO requires opsonin receptors (FcR and complement receptors), scavenger receptors or Toll-like receptors. EXO can bind with opsonins and are internalized by the cell where they fuse with limiting membrane of endocytic compartments. Most EXO are sorted to endosomal rather than lysosomal pathways. Phagocytosis of EXO is dependent on the actin cytoskeleton, PI3K, and dynamin2. Phagocytosis can be blocked by using actin polymerization inhibitor as cytochalasin D [95].

#### 7.4.2. Pinocytosis

Macropinocytosis. During macropinocytosis, plasma membrane protrusions are driven by polymerization of actin filaments forming an invagination for non-specific endocytosis of extracellular fluids and small particles. Phosphatidyl serine on the surface of EXO activates macropinocytosis. Macropinocytosis of EXO is dependent on Na+/H exchange, actin and PI3K, it can be inhibited using of Na+/H+ ion exchange inhibitors such as amiloride [96,97].

Micropinocytosis includes clathrin dependent endocytosis (ligand/receptor-mediated endocytosis) and clathrin independent endocytosis (lipid raft-mediated endocytosis). The former involves EXO ligand engagement with specific receptors on the cellular plasma membrane. It utilizes clathrin and adaptor protein 2 complexes which coat the membrane and induce invagination of the membrane into a vesicle. It is also dynamin dependent and can be inhibited by clathrin knock down using siRNAs [86,98]. The latter involves dynamin dependent and dynamin independent mechanisms, requiring cholesterol and sphingomyelin-rich microdomains in the plasma membrane. Dynamin dependent is mediated by caveolin 1) and/or RhoA kinase which can be inhibited by siRNA knock down of caveolin 1 or RhoA kinase respectively. Dynamin independent is divided into ARF6 or CDC42 GTPase and Flotilin-1 mediated. Studies showed knock down of flotillin as well as inhibitors or silencing the associated GTPases inhibits endocytosis by this mechanism [86,98].

## 8. Exogenously Produced DC EXO: Natural Nano-Delivery Systems for Inflammatory Diseases

The therapeutic advantages of EXO nano-delivery systems have been previously reviewed in detail [14,35,99,100,101,102,103,104,105]. DC-derived EXO mimic the biology of donor DCs generating considerable interest in their use as cell free therapeutic agents [106]. During interaction with cognate T cells, DCs promote the biogenesis of MVPs with MHC and CD9, carrying intraluminal vesicles to be exocytosed upon fusion with the plasma membrane [65,107]. Mature immune-stimulatory (mDCs EXO) or immature immune regulatory (imDCs EXO) can be isolated from mature or immature DCs, respectively. mDCs EXO loaded with specific peptides can elicit potent specific immune activation, resulting in tumor cell detection and eradication, and microbial elimination [22,108,109]. In clinical trials, autologous mDCs EXO vaccinated into cancer patients showed additional therapeutic effects without significant side effects [110,111]. EXO from DCs can be pulsed with bacterial antigen peptides for bacteria-free vaccines [112,113] or with *Toxoplasma gondii* anti-parasite immunity in mice [114].

On the opposite side of the immune spectrum, immune tolerance can be induced by iDC EXO. DCs modified with anti-inflammatory molecules, including IL-10, IL4, TGFB, FasL, IDO, and CTLA4 have generated various immunoregulatory EXO subtypes for treatment of immune disorders [14,93,99,100,102,103,104,105]. DC lineage marker CD11c and low levels of antigen presenting MHCII and costimulatory molecules CD80, CD86 are a feature of EXO from immature or tolerogenic DCs. iDC EXO inhibit T cell activation and proliferation directly or indirectly by reprograming the biological function of acceptor DCs and promoting a regulatory/suppressive subset [14,93,103]. Delivery of iDC EXO by an intradermal route resulted in their interaction with acceptor CD11c+ cells in dermis and regional lymph nodes. Macrophages and CD11c+ DCs in liver and spleen were found to take up EXO injected intravenously [103]. iDCs EXO carrying tolerogenic molecules inhibit rheumatoid arthritis and joint destruction in animal models [14,102,103,104]. In an inflammatory bowel disease (IBD) model, TGF-β1 enriched iDCs EXO reversed the disease severity and clinical indices. This occurred through repression of Th17 responses and enhancement of T-regulatory cells (Tregs) [93]. Experimental autoimmune encephalomyelitis responded well to injection of iDCs EXO with membrane bound TGF-β1 [115]. iDC EXO can prolong the survival times of cardiac and liver transplantation [116,117] and improve cardiac function in mice post MI [118]. Our team recently found that immune-regulatory EXO (regDCs EXO) subsets, loaded with TGFb1 and IL10, were retained at the inflammatory sites of oral injection and were efficient in reprogramming the immune function of local acceptor DCs and CD4+ T cells, inducing tolerogenic DCs and Tregs in situ. Moreover, regDC EXO protected their therapeutic cargo against proteolytic degradation and abrogated inflammatory bone loss in experimental periodontitis [119] (Figure 1). The proteomic cargo and biodistribution characteristics of DC EXO were further examined. The protein cargo of regDC EXO was indeed complex, but dominated by tissue trafficking, cell binding, and immunoregulatory proteins were dominant, consistent with their functions [120]. Interestingly intravenous injection of regDCs EXO in mice led to predominant accumulation in the lungs. This generated interest in possible therapeutic application of regDC EXO for COVID-19. We showed that regDCs EXO inhibit the expression of ACE2, the SARS-CoV-2 target receptor, in respiratory tract epithelial cells (PBTECs) by a TGFb1 dependent mechanism. This generated speculation that regDC EXO may have efficacy in reversing harmful inflammatory lung responses in COVID-19, as reported [119]. Another approach is to block the SARS-CoV-2 entry point using regDC EXO therapy, reducing the infection severity [121]. Other suggested approaches involve EXO enrichment with antiviral drugs [122], decoy ACE2 [123], or the S-protein of the SARS-CoV-2, for COVID-19 infection [124].

EXO have been isolated from other immune cells for immune reprogramming, including Tregs [125,126,127] B lymphoblasts [128,129], natural killer cells [130] and mast cells [131]. EXO from macrophages were loaded with linezolid and vancomycin to inhibit intracellular infection by methicillin-resistant Staphylococcus aureus (MRSA) [132,133]. Other groups loaded EXO derived from muscle cells with various virus antigens to be used as a vaccine and induce specific cytotoxic T lymphocyte immunity [134]. In summary, DC EXO are preferable to whole immune cell-based therapy for inflammatory and infectious diseases for many reasons [14,35,99,100,101,102,103,104,105].

## 9. Endogenously Produced DC EXO: Diagnostic and Pathogenic Potential, Role in Immune Senescence

### 9.1. EXO Diagnostics

Endogenously produced EXO are found in the intracellular spaces, in tissues and body fluids. The proteins, lipids, and nucleic acids [135] in EXO are being studied for defining underlying diseases and conditions. Saliva EXO for example, are under intensive study as a non-invasive diagnostic of premalignant cancer lesions [136]. Salivary EXO from periodontitis patients is enriched in PDL-1 and miRNAs, hsa-miR-140-5p, hsa-miR-146a-5p, and hsa-miR-628-5p) compared with healthy controls [137,138]. EXO and their biologically active cargos offer diagnostic and prognostic information in chronic airway diseases [139], craniofacial disorders [140], renal diseases [141,142,143], neurodegenerative diseases [144], lipid metabolic diseases [145], and tumors [146].

### 9.2. EXO Pathogenesis

The potential pathogenic role of endogenous DC EXO cannot be overlooked since it can provide additional information about therapeutic applications. During HIV infection, DCs release HIV-bearing EXO that serve as a shuttle carrying HIV viral particles and transmitting infection to CD4T cells [147,148]. Fibronectin and galactin 3 may be instrumental in transmission of HIV infection to bystander T cells by DC EXO [149]. These viral-bearing DC EXO are more infectious and less efficiently eradicated than cell-free viral particles [150].

Involvement of DC EXO has been documented in chronic inflammatory diseases and allergic reactions [119,151,152,153]. We have recently shown the role of EXO derived from mature DCs in exacerbating an inflammatory response and alveolar bone loss in a murine model of periodontitis [119]. Further, ongoing unpublished work from our laboratory has revealed that EXO derived from DCs infected with the keystone oral pathogen *P.gingivalis*, aggravates inflammatory bone loss in mice. Mature DC EXO migrate to the spleen via CCR7 and induce inflammation in vivo [154] and exacerbate atherosclerosis through TNFa/Nf-KB-mediated stimulation of endothelial cells [155]. This study showed a significant increase in atherosclerotic lesions in APOE^−/−^ mice upon 12-week injection of mature DC EXO [155]. Similarly, EXO secreted from atherogenic macrophages were found to promote atherosclerosis. Transfer of specific EXO miRNAs to recipient macrophages inhibits migration and promotes their entrapment in the blood vessel wall [156]. On the other hand, DC EXO have demonstrated a protective effect in myocardial infarction (MI) [157,158] and Ischemic reperfusion injury (IRI) [158], potentially through induction of Tregs. DC EXO contribute to the production of leukotrienes LT and other inflammatory mediators of arachidonic acid pathway [151,152]. Moreover, DC EXO can acquire and present the aeroallergen, Fel d1 inducing a Th2 response with the production of IL4 in PBMCs from individuals with cat allergy [153]. Additionally, donor DC EXO elicit an immune reaction and promote allograft rejection in cardiac transplantation in mice [159]. Serum EXO from HCV-infected patients contains HCV RNA and induces transmission of HCV to liver cells through viral receptor-independent mechanisms [160]. Serum EXO isolated from patients with tuberculosis were found to carry mycobacterial antigens [161]. EXO of intestinal lumen aspirates of inflammatory bowel disease IBD patients carry pro-inflammatory factors including TNF-α, IL6, and IL8 in levels higher than that of healthy controls. These EXO activate epithelial cells and macrophages to secrete IL-8 [162]. EXO purified from synovial fibroblast of rheumatoid arthritis patients contain TNF-α at the transmembrane domain that induces NF-κB activation and MMP-1 release in acceptor bystander cells [163]. DC EXO have been used as prognostic biomarkers for metastatic melanoma treated with Ipilimumab reflected by the level of CD86 expression [164].

### 9.3. Alteration of Exosome Secretion and Cargo Content in Infection

Pathogenic invasion of the host cell changes its metabolic activity and endocytic pathway utilization, which in turn affects EVs secretion [71,165]. An increase in the host cell secretion of EXO and associated protein was observed in *Plasmodium*, Rota virus, *Chlamydia psittaci*, and *Mycobacterium bovis* Bacille Calmette–Guerin (M.bovis BCG) infections [165,166,167,168,169]. Moreover, the concentration of secreted EVs in gingival crevicular fluid (GCF) was reported to increase in periodontitis patients compared to healthy/gingivitis subjects [170]. In line with these studies, our group has recently shown that *P.gingivalis* infection of host DCs leads to an increase in the number of secreted EXO [71], which was restored to base line levels with rapamycin treatment, a known inducer of autophagy [71], thus suggesting a potential role of autophagy in EXO secretion during infection. Autophagy inhibition promotes an increase in EXO secretion in response to accumulation of damaged proteins or organelles, whereas activation of autophagy diminishes EXO release due to the fusion of MVB and autophagosomes [71,171]. Other mechanisms for enhanced release of infection-mediated EXO have been discussed [165]. Ebola and HIV viral infections activate EXO biogenesis by increasing synthesis of CD63, apoptosis-linked-gene-2 product-interacting protein X (Alix) and endosomal sorting complex required for transport machinery-II (ESCRT) proteins [165,172,173,174]. In addition to an increase in the number of released EXO during infection, the host cell-derived EXO exhibit a distinct proteomic, lipid and nucleic acid profile. In this context, the microRNA (miRNA) content of DC EXO, is worth mentioning, as it results from a complex sorting process during cellular biogenesis and export [60]. MiRNAs are especially active in post transcriptional regulation of genes [175]. Recently, we have shown a change in miRNA content of EXO from DCs infected with *P. gingivalis* [71], contributing to inhibition of beclin-mediated autophagy [176], suppression of ULK1 and LC3I/II [177] (miR17-5p), and impediment of autophagy-dependent clearance of intracellular bacteria [178] (miR106B). Further, our study showed upregulation of miRNA 132-3p in Pg-induced DC EXO, consistent with aged bone-marrow derived DCs (BMDC). Infection-induced DC EXO often contain pathogen-related proteins [71,108,109] and virulence factors [71,179,180,181] that can affect the physiologic host response in multiple ways [165]. EXO can transmit infections to bystander cells [147,172], can evade the immune response [147,150], induce cell apoptosis [165,180,182] or promote immune suppression [113,165]. On the contrary, antigen-bearing EXO can activate the immune response and illicit an antigen specific T cells response [165,183]. Moreover, the efficacy of DC EXO, pulsed with *Toxoplasma Gondii*, *Chlamydia psittaci*, *sporozoites*-extracted- or tumor antigens to elicit an immune response has highlighted DC EXO as a strong candidate for cell free vaccines [108,109,169]. Worthy of note, is the ability of DC EXO to activate a humoral antibody response when challenged with Diphtheria Toxoid, or streptococcus pneumoniae [179,184]. The immunogenicity of EXO from other antigen presenting cells have been reported. For example, *Mycobacterium tuberculosis* or *Mycobacterium bovis*-infected macrophages secrete EXO bearing bacterial antigens to stimulate antigen specific immune responses [112,185].

### 9.4. Role of DC EXO in Immune Senescence and Inflammaging

A well-established feature of physiological ageing is diminished immune function, or immune senescence, coupled with a low-grade chronic inflammation [186,187]. Senescence is regulated by several signaling pathways, most notably, Akt1 [188,189] and mTOR [190,191], with the latter inhibited by rapamycin [191]. Functional deficits are reported in senescent monocytes, macrophages (reviewed in [192]), and T cells (reviewed in [193]). Vulnerability of DCs to senescence was attributed to cellular stressors such as microbial infection [71]. The senescence associated secretory phenotype (SASP), a distinctive feature of senescence (Figure 2), consists of secreted exosomes (EXO) and inflammatory cytokines [194,195,196,197]. Mounting evidence indicates a critical role of EXO in immune senescence, inflammaging and age-related diseases (ARDs) [198]. Reports have shown that senescence leads to an increase in EXO secretion [199] (Figure 2). Recently, we have shown that exosomes derived from *P.gingivalis*-infected DC are increased in number, transmit immune senescence to bystander immune cells in vitro [71], and accelerate alveolar bone loss in a PD model in vivo (unpublished data). Further, *P. gingivalis*-induced DC EXO comprised SASP-related molecules, TNFa, IL1B and IL6, and Mfa1 fimbrial protein [71]. Prior published work from our group using human monocyte derived DCs (MoDCs) [200,201,202,203,204,205,206] has identified an important role for Pg minor Mfa1 fimbriae in invasion of MoDCs, activating mTOR (inhibiting autophagy) and hyper-phosphorylating Akt1 (inhibiting apoptosis), suggesting a role in premature immune senescence [189]. An increased role for Mfa1 fimbriae in vivo is suggested by its upregulation in Pg from dental plaques and blood DCs of humans with PD [207]. Activation of the inflammasome, implicated in periodontal inflammaging [208] reportedly orchestrates the SASP [194]. Microbial activation of host DCs triggers inflammasome activation [209] and secretion of pathogenic exosomes that contain mature IL-1B [71]. These pathogenic EXO transmit immune senescence to normal bystander cells, amplifying senescence in paracrine [71]. A study of immune senescence has yielded a wealth of knowledge about ARDs [193]. However, much work is required to better understand the role of EVs in senescence and age-related diseases.

## 10. Extracellular vesicles (EV), DC EXO and Cancer

Several studies have shown the role of EVs in tumorigenesis. Tumor cells were found to secrete EVs that bear immunosuppressive molecules and disrupt differentiation, maturation, and function of DC [210]. Moreover, tumor EVs inactivate T lymphocytes or natural killer cells to suppress immune responses, promoting tumor progression [211]. At the site of the primary tumor, tumor cells secrete fibronectin-coated EXO at the leading edge of the cell that bind collagen fibrils extracellularly while interacting with integrins on cell membrane. This enhances cancer cell motility and migration by stabilizing cellular protrusions and facilitates amoeboid movement [212]. EXO-carrying multivesicular bodies are docked at cellular protrusion called invadopodia and secrete growth factors and matrix metalloproteinases containing EXO that enhance invadopodia stability, extracellular matrix (ECM) degradation, and invasion. [213,214]. On the other hand, tumor-derived EXO harbor tumor antigens that can activate DCs, stimulating antitumor specific cytotoxic immune response. This is more effective than that achieved by tumor cell lysates or soluble-free antigens when used as vaccines [215]. In addition, DCs EXO have been employed as a subcellular vaccine for cancer therapy in clinical trials [110,111,216] (Table 1). These include two phase I [110,111] and one phase II [216] clinical trials in end-stage cancer patients. The first phase I clinical trial utilized EXO isolated from autologous immature monocyte derived DCs (MoDCs). These DC EXO were loaded with both MHCI and MHCII melanoma associated antigen (MAGE) peptides. Four DC EXO vaccinations were administered at weekly intervals into 13 advanced non-small cell lung cancer (NSCLC) patients. Nine of these patients completed the therapy. Limited DTH reactivity against MAGE peptides, MAGE-specific T cell responses with increased natural killer cell (NK) lytic activity were observed. In addition, DC EXO therapy was found to be safe and well-tolerated [111]. These DCs EXO were also employed in the second phase I clinical trial once weekly for 4 weeks in 15 metastatic melanoma patients. NK cell functions were augmented and only grade 1 toxicity was observed [110]. Phase II clinical trial utilized EXO derived from IFN—stimulated autologous MoDCs. The EXO were loaded with both MHCI and MHCII restricted tumor antigens and administrated in non-small cell lung cancer in one-, two- and three- week intervals in a maintenance immunotherapy protocol. NK cell activity was also promoted in phase II clinical trial with limited T cell activity. Twenty two of the twenty-six patients completed the study. Grade 3 hepatotoxicity was found in one of the cases [216]. These phase I and II clinical trials showed the feasibility of DC EXO as anticancer vaccines, suggesting the safety and potentiality for other systemic diseases.

## 11. Conclusions

Exogenously produced exosomes from dendritic cells, custom tailored to activate, suppress or reprogram the immune system are under intensive study for a variety of infectious or inflammatory diseases, including periodontitis. The efficacy and safety of these therapeutic nanoparticles are predicated on further in-depth study of their biogenesis, biodistribution, and modes of cell-to-cell communication. Careful consideration must also be given to the diagnostic and pathologic significance of endogenously produced exosomes, secreted at inflammatory sites, into tissues and body fluids. The molecular cargo of such endogenous exosomes, including infectious agents, proteins, lipids, and nucleic acids, must be thoroughly studied to understand their role in disease, but also their potential for transmitting such cargo, in paracrine, to neighboring cells. This is particularly relevant to transmission and amplification of immune senescence between immune cells with or without cell-to-cell contact.

## Figures and Tables

**Figure 1 cells-11-00115-f001:**
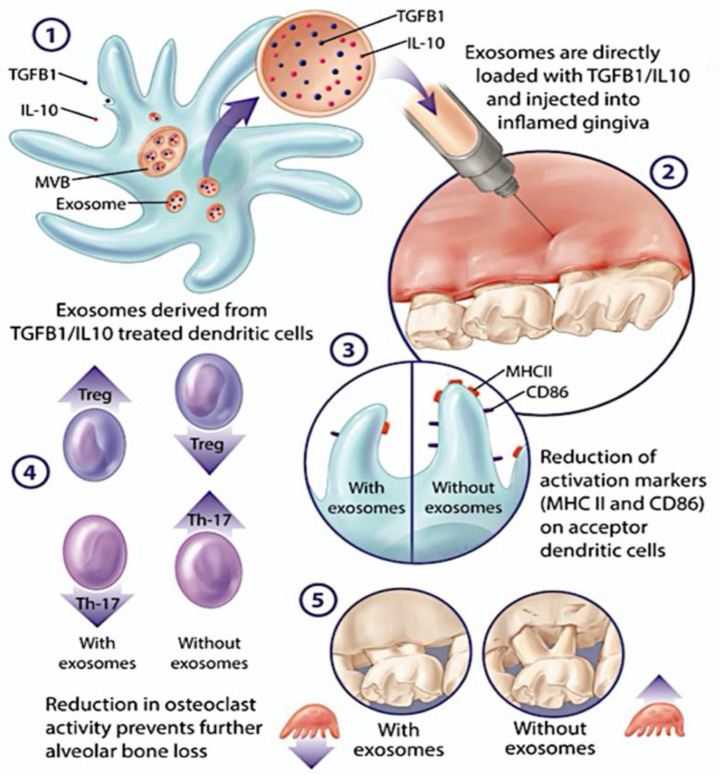
DC Exosome (Exo) Therapy for the Inflammatory Bone Disease Periodontitis. 1. TGFb-/IL-10 enriched exo (regDC exo) are released. 2. Exo injected into palatal gingiva interproximal at site of ligand induced PD. 3. Exo uptake by acceptor gingival DCs reduces DC maturation and alters cytokine profile; 4 and 5. Modulation of osteoclastogenic Th17 response by regDCexo.

**Figure 2 cells-11-00115-f002:**
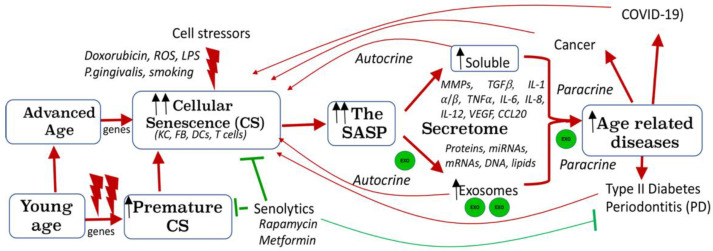
Cellular Senescence due to Advanced Age and Cell Stressors: Role of Exosomes. Advanced age and canonical (e.g., doxorubicin) and non-canonical (e.g., P.gingivalis) cell stressors can provoke cellular senescence (CS). Premature CS can occur by exposure to X irradiation, doxorubicin, reactive oxygen species (ROS), and microbial CS stressors. Senolytic agents (e.g., rapamycin, metformin) can remove senescence cells. CS profiling identifies elevated SA-β-Gal, p16 INK4A, pAkt-1, p53, p21Waf1/Clip1, and soluble and exosomal SASP. Exosomes (exo, green circles) can transmit senescence to young cells in paracrine. CS is implicated in many age-related diseases such as periodontitis, type II diabetes and COVID-19.

**Table 1 cells-11-00115-t001:** DC EXO Clinical Trials.

Disease Type	Phase	*n*	DCs EXO	Doses	Outcome	Ref
Advanced Non-small cell lung cancer	I	13 (9 completed the study)	Autologous MoDCs derived EXO were loaded with MAGE peptides.	4 vaccinations at weekly intervals.	Limited T cell reactivity and DTH against MAGE peptides. Increased NK lytic activity. Safe, well tolerated activity.	[186]
Metastatic Melanoma	I	15	Autologous MoDCs derived EXO were loaded with MAGE peptides.	4 vaccinations at weekly intervals.	No MAGE-specific T cell response. No DTH response. NK cell activation. Safe, well tolerated.	[187]
Advanced Non-small cell lung cancer	II	26 (22 completed the study)	EXO were isolated from IFN-stimulated autologous MoDCs and loaded with MHCI and MHCII restricted cancer antigens.	Vaccination in 1, 2 and 3 week intervals in a maintenance immunotherapy protocol.	Limited T cell activity. increased NK cell function. One patient had a grade-3 hepatotoxicity.	[188]

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
