# Peer review of "Exogenous and Endogenous Dendritic Cell-Derived Exosomes: Lessons Learned for Immunotherapy and Disease Pathogenesis"

_cells, 2021, doi:10.3390/cells11010115_

Round 1

Reviewer 1 Report

The review is dedicated to the the physiology of DC-derived exosomes and their role in disease and treatment.

The abstract is concise and the review is thorough and well written and gives a general view on exosomes, dendritic cells and their role in immune response. 

The figures are masterpiece and very well put in the text.

The aspects of DC-derived exosomes in cancer are missing.  I would ask the authors to dedicate a separate paragraph to this topic. 

Also please expand on the pharmacological use of IV exosomes with an added table on the clinical trials up to date.

Author Response

The suggestion of a section on cancer, with a table  is a good one.   This has been added now in the revised version at page 13, section 10. The new table is on page 14.  

Reviewer 2 Report

Exogenous and endogenous dendritic cell-derived exosomes: lessons learned for immunotherapy and disease pathogenesis" from Elashir M and el. provides to the reader a quick update on an emerging field of immunobiology, while discussing interesting translational perspectives. The review is interesting for a wide audience and well written.

Author Response

We appreciate the reviewers comments.

Reviewer 3 Report

In this review the authors summarize the current knowledge about dendritic cell-derived exosomes endogenously or exogenously produced. They also describe the general features of exosomes but this part is not well organized and it should be modified or improved. In particular, the paragraphs 4,5,6 and 7 are divided in very short parts and don't include a detailed description of all the concept included (in particular in the paragraphs 5 and 6). Moreover, in the sub-paragraph 4.3.3 the authors should describe the specific mechanism of loading of miRNAs in exosomes, as those described in these studies: Villaroya-Beltri et al., Nature Commun. 2013 or Santangelo et al., Cell Rep. 2016.

Author Response

The reviewer provides some astute suggestions that will greatly improve the review. These have now been addressed in the revised version, tracked on pages 3, 5, 6, 7